# Modulation of the morphotropic phase boundary for high-performance ductile thermoelectric materials

Jiasheng Liang[1,2,5], Jin Liu[1,2,5], Pengfei Qiu [1,2,3] ✉, Chen Ming [1], Zhengyang Zhou[1], Zhiqiang Gao[1,4], Kunpeng Zhao [4] ✉, Lidong Chen [1,2] & Xun Shi [1,2,4] ✉

The flexible thermoelectric technique, which can convert heat from the human body to electricity via the Seebeck effect, is expected to provide a peerless solution for the power supply of wearables. The recent discovery of ductile semiconductors has opened a new avenue for flexible thermoelectric technology, but their power factor and figure-of-merit values are still much lower than those of classic thermoelectric materials. Herein, we demonstrate the presence of morphotropic phase boundary in $Ag_2Se$-$Ag_2S$ pseudobinary compounds. The morphotropic phase boundary can be freely tuned by adjusting the material thermal treatment processes. High-performance ductile thermoelectric materials with excellent power factor ($22\,\mu Wcm^{-1}K^{-2}$) and figure-of-merit (0.61) values are realized near the morphotropic phase boundary at 300 K. These materials perform better than all existing ductile inorganic semiconductors and organic materials. Furthermore, the in-plane flexible thermoelectric device based on these high-performance thermoelectric materials demonstrates a normalized maximum power density reaching $0.26\,Wm^{-1}$ under a temperature gradient of 20 K, which is at least two orders of magnitude higher than those of flexible organic thermoelectric devices. This work can greatly accelerate the development of flexible thermoelectric technology.

Flexible thermoelectrics can directly convert heat from the human body to electricity based on the Seebeck effect with the advantages of better portability, higher reliability, and longer lifetime than chemical batteries, making it a highly promising self-powered technology for wearable devices[1–4]. High-efficiency flexible thermoelectric (TE) technology requires materials with both high TE performance and good flexibility to adapt to curved body surfaces. The performance of a TE material can be evaluated by the dimensionless figure of merit ($zT = \alpha^2\sigma T/\kappa$), where $\alpha$ is the Seebeck coefficient, $\sigma$ is the electrical conductivity, $\kappa$ is the thermal conductivity, and $T$ is the absolute temperature[5–10]. In particular, high power factors ($PF = \alpha^2\sigma$) are essential for providing large output power under an assigned temperature difference ($\Delta T$).

To achieve excellent flexibility, organic conducting polymers (e.g., PEDOT, P3HT, and PANI) are natural candidates for flexible thermoelectrics due to their inherent soft, bendable, and ductile

---

[1]State Key Laboratory of High Performance Ceramics and Superfine Microstructure, Shanghai Institute of Ceramics, Chinese Academy of Sciences, Shanghai, China. [2]Center of Materials Science and Optoelectronics Engineering, University of Chinese Academy of Sciences, Beijing, China. [3]School of Chemistry and Materials Science, Hangzhou Institute for Advanced Study, University of Chinese Academy of Sciences, Hangzhou, China. [4]State Key Laboratory of Metal Matrix Composites, School of Materials Science and Engineering, Shanghai Jiao Tong University, Shanghai, China. [5]These authors contributed equally: Jiasheng Liang, Jin Liu. ✉e-mail: qiupf@mail.sic.ac.cn; zkp.1989@sjtu.edu.cn; xshi@mail.sic.ac.cn

mechanical properties[11–13]. However, their carrier mobilities are extremely low (usually less than $1\,cm^2V^{-1}s^{-1}$), resulting in lower $PF$ (usually less than $1\,\mu W\,cm^{-1}K^{-2}$) and $zT$ values (usually less than 0.1) than classic brittle inorganic TE materials with high $PF$ (higher than $20\,\mu W\,cm^{-1}K^{-2}$) and $zT$ (approximately 0.8-1.0) values at room temperature[14,15]. The recently discovered ductile inorganic materials, such as Ag$_2$S-based compounds, are highly suitable for flexible thermoelectrics because they possess mechanical properties similar to those of organic polymers and high carrier mobilities similar to those of classic inorganic materials[1,16–19]. The former can lead to excellent flexibility, while the latter can result in high $PF$ and $zT$ values similar to those of inorganic TE materials. At room temperature, Ag$_2$S is ductile, but its $PF$ and $zT$ values are extremely low (negligible at 300 K)[17]. Alloying a certain amount of Se/Te in Ag$_2$S leads to the discovery of a series of n-type and p-type high-performance ductile inorganic TE materials with a maximum $PF$ value of ~5 $\mu W\,cm^{-1}K^{-2}$ and a maximum $zT$ value of ~0.45 at 300 K[1,17,19–28]. These numbers are superior to those of present organic TE materials[29–35], but they are still much lower than those of classic Bi$_2$Te$_3$- and Ag$_2$Se-based TE materials[14,15,36–38], greatly limiting the power output and applicability of flexible devices based on these ductile TE materials.

The morphotropic phase boundary (MPB) is a transition region in the oblique temperature–composition phase diagram that separates two competing phases with different crystallographic symmetries[39–43], such as the rhombohedral and tetragonal structures in lead zirconate titanate (PZT). The sharp extrema in piezoelectric properties near the MPB lead to unprecedented piezoelectric properties from both fundamental research and practical applications[39,40]. However, MPB has rarely been observed in TE materials due to the lack of material competing phases with distinct crystallographic symmetry and TE performance characteristics. In Ag$_2$(Se,S)-based compounds, the crystal structure changes from orthorhombic to monoclinic with increasing S content. This transformation indicates that an oblique temperature–composition phase boundary may exist. More importantly, both the mechanical ductility and TE performance are quite different between orthorhombic and monoclinic materials; that is, orthorhombic Ag$_2$Se is fragile and has a high $PF/zT$ ($zT$ reaching 1.0) at room temperature, while monoclinic Ag$_2$S is ductile and has a low $PF/zT$ at room temperature[14,17,44]. All these features strongly indicate that MPBs similar to those in PZT-based compounds may exist in Ag$_2$(Se,S)-based compounds.

In this work, we demonstrate that the MPB separating orthorhombic and monoclinic structures is present in Ag$_2$Se-Ag$_2$S pseudobinary compounds. This boundary can be freely tuned by adjusting the thermal treatment process. The orthorhombic Ag$_2$Se$_{1-x}$S$_x$ pseudobinary compounds near the MPB effectively integrate the characteristics of excellent TE performance and good ductility, boosting the maximum room-temperature $PF$ and $zT$ values of ductile materials to $22\,\mu W\,cm^{-1}K^{-2}$ and 0.61, respectively (Fig. 1). Furthermore, based on these high-performance ductile TE materials, a fully flexible in-plane TE device is successfully developed with a superior normalized material exhibiting a maximum power density reaching $0.26\,W\,m^{-1}$ under a temperature gradient of 20 K.

## Results and discussion

When the S content $x$ is smaller than 0.2, Ag$_2$Se$_{1-x}$S$_x$ pseudobinary compounds have orthorhombic structures similar to those of Ag$_2$Se; when $x$ is larger than 0.4, Ag$_2$Se$_{1-x}$S$_x$ pseudobinary compounds have monoclinic structures similar to those of Ag$_2$S (see the X-ray diffraction patterns obtained for bulk Ag$_2$Se$_{1-x}$S$_x$, Fig. S1)[45]. Accordingly, the mechanical and TE properties are determined by the crystal structures of the material. When $x \leq 0.2$, the materials are brittle with high $PF$ and $zT$ values similar to those of Ag$_2$Se; when $x \geq 0.4$, the materials are ductile with low $PF$ and $zT$ values similar to those of Ag$_2$S[44].

The case is quite different and complicated when the S content $x$ is between 0.2 and 0.4. By taking Ag$_2$Se$_{0.61}$S$_{0.39}$ as an example, the heat flow curves measured by differential scanning calorimetry (DSC) are displayed in Fig. 2a and Fig. S2. Differing from the single endothermic peak observed in Ag$_2$S (monoclinic–cubic transition) and Ag$_2$Se (orthorhombic–cubic transition), there are two endothermic peaks from 300 K to 400 K in Ag$_2$Se$_{0.61}$S$_{0.39}$, indicating that there are two phase transitions. The powder X-ray diffraction patterns collected at different temperatures indicate that Ag$_2$Se$_{0.61}$S$_{0.39}$ crystalizes in an orthorhombic structure at 300 K, in a monoclinic structure at 350 K, and in a cubic structure at 400 K (Fig. 2b). The Rietveld refinement proves that the orthorhombic phase and monoclinic phase are both pure (Fig. S3 and Table S1). Thus, the two-phase transitions are an orthorhombic–monoclinic transition and a monoclinic–cubic transition. Furthermore, the phase transition temperatures determined by the measurements during the heating and cooling processes are obviously different, indicating strong thermal hysteresis.

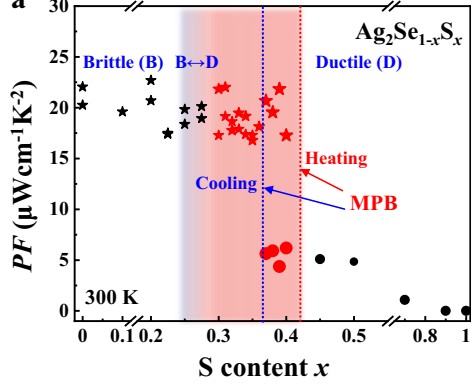
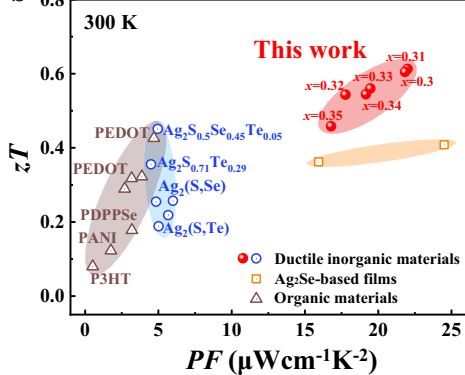

**Fig. 1 | High-performance ductile Ag$_2$Se$_{1-x}$S$_x$ pseudobinary compounds.**
**a** Room-temperature power factor ($PF$) as a function of $x$ in Ag$_2$Se$_{1-x}$S$_x$ pseudobinary compounds. The dashed lines represent the orthorhombic–monoclinic morphotropic phase boundary (MPB) obtained during heating and cooling. The pentagrams represent orthorhombic structures, and the circles represent monoclinic structures. The red pentagrams and circles represent ductile Ag$_2$Se$_{1-x}$S$_x$ near

the MPB. **b** Room-temperature TE figure of merit ($zT$) as a function of $PF$ for ductile orthorhombic Ag$_2$Se$_{1-x}$S$_x$ ($0.30 \leq x \leq 0.35$) pseudobinary compounds. Previously reported data about other typical ductile inorganic TE materials, organic TE materials, and Ag$_2$Se-based films are included for comparison. The data are taken from Refs. 1,17,19,20,22,29–38,44.

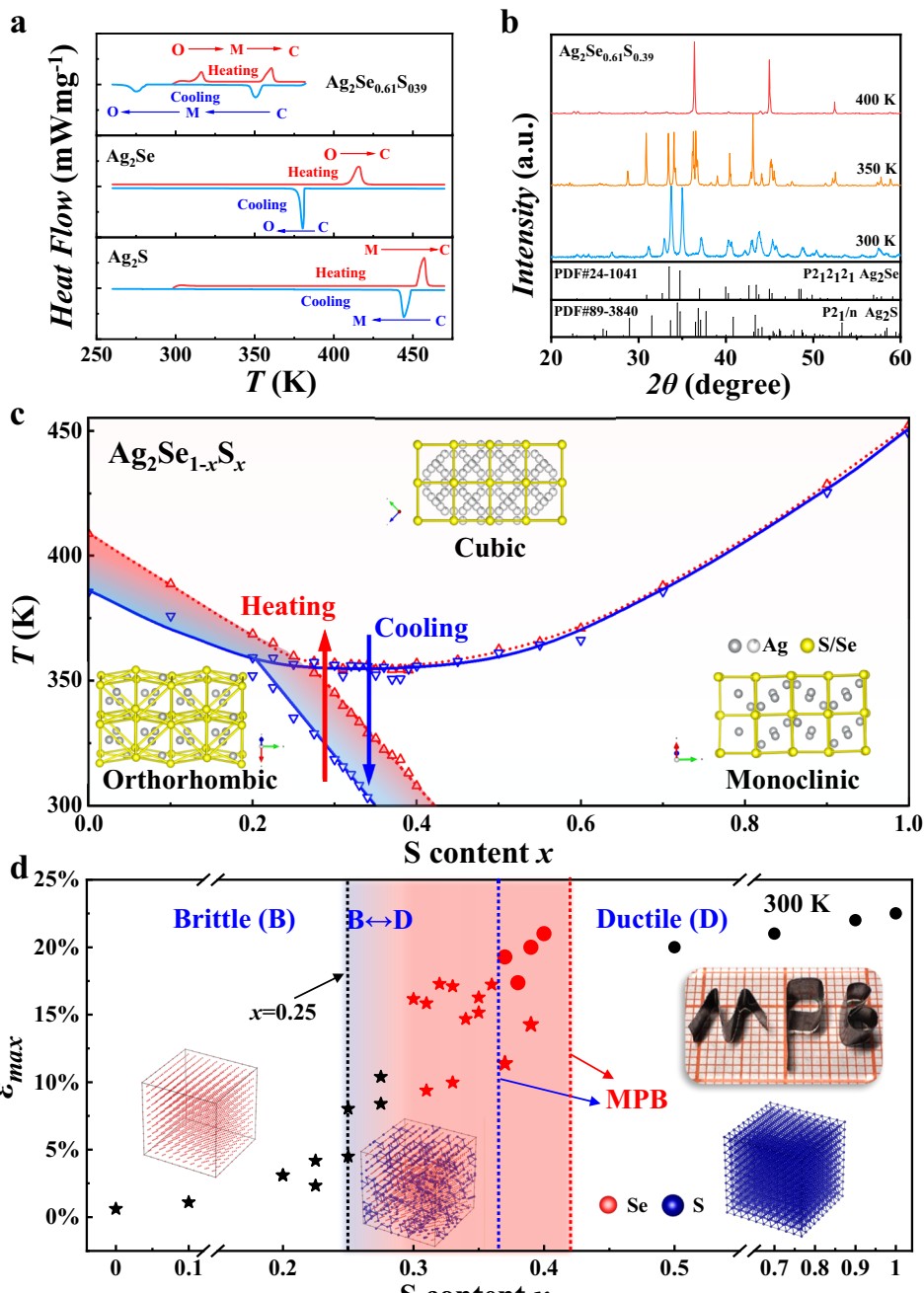

**Fig. 2 | Phase transition and mechanical properties of $Ag_2Se_{1-x}S_x$ pseudobinary compounds. a** Heat flow curves of $Ag_2Se_{0.61}S_{0.39}$ during heating and cooling processes. The letters O, M, and C represent orthorhombic, monoclinic, and cubic structures, respectively. **b** Powder X-ray diffraction patterns of $Ag_2Se_{0.61}S_{0.39}$ measured at different temperatures. The powders are ground in liquid nitrogen. **c** $Ag_2Se$-$Ag_2S$ phase diagram between 250 and 480 K based on the heat flow curves shown in Fig. S2. The red and blue lines represent the phase boundaries in the heating and cooling processes, respectively. The triangles represent the experimental data. The insets in **c** are schematics of the cation Se/S framework in orthorhombic, monoclinic, and cubic structures. **d** Maximum bending strain ($\varepsilon_{max}$) as a function of $x$. $\varepsilon_{max}$ is determined by the three-point bending test. The brittle–ductile boundary occurs at approximately $x = 0.25$–$0.30$. The upper right inset is an optical image of orthorhombic $Ag_2Se_{0.67}S_{0.33}$ twisted into different shapes. The bottom insets show the schematics of the percolation effect in three dimensions, with the percolation threshold occurring at approximately $x = 0.25$–$0.31$. The stars and circles represent orthorhombic and monoclinic $Ag_2Se_{1-x}S_x$, respectively. The red pentagrams and circles represent ductile $Ag_2Se_{1-x}S_x$ near the MPB.

By using the same method, a $Ag_2Se$-$Ag_2S$ pseudobinary phase diagram between 300 and 480 K is obtained (see Fig. 2c). The red lines represent the phase boundaries in the heating process, while the blue lines represent the phase boundaries in the cooling process. When $x$ is larger than 0.4, the phase transition temperatures (monoclinic–cubic transition) are almost the same between the heating and cooling processes. When $x$ is smaller than 0.2, the phase transition temperatures of orthorhombic–cubic transitions in the cooling process are lower than those in the heating process. When $x$ is 0.2–0.4, the phase transition temperatures of the orthorhombic–cubic transitions and orthorhombic–monoclinic transitions in the cooling process are lower than those in the heating process. This thermal hysteresis leads to different phase boundaries, as shown in Fig. 2c. By taking $Ag_2Se_{0.61}S_{0.39}$ as an example, the room-temperature structure is orthorhombic in the heating process and monoclinic in the cooling process (Fig. S4). Nevertheless, the orthorhombic–monoclinic transition temperature

gradually decreases with increasing $x$, resulting in oblique temperature–composition phase boundaries similar to those of the well-known MPB between the tetragonal and rhombohedral PZT phases[39,40]. The cubic $Ag_2Se_{1-x}S_x$ ($0.2 < x < 0.4$) pseudobinary compounds are first converted to a monoclinic structure and then converted to an orthorhombic structure in the cooling process; these results are reversed in the heating process.

These composition-induced orthorhombic–monoclinic transitions can be understood by the energy variations using first-principle calculations. The formation energies of the materials with orthorhombic structure ($E_O$) and monoclinic structure ($E_M$) are shown in Fig. S5. The differences ($E_O$–$E_M$) are 19.7 meV atom⁻¹ when $x=0$ and −27.1 meV atom⁻¹ when $x=1$, consistent with the scenario in which $Ag_2S$ crystalizes in a monoclinic structure, while $Ag_2Se$ crystalizes in an orthorhombic structure at room temperature. The difference ($E_O - E_M$) increases with increasing $x$, reaching zero when $x$ is approximately 0.6 (Fig. S5). The structural competition between these two phases with nearly the same energies is responsible for the formation of MPB in the experimental phase diagram, as shown in Fig. 2c.

Interestingly, although the lattice symmetry of the orthorhombic structure is higher than that of the monoclinic structure, the cation Se/S framework in the cubic structure is more similar to that in the monoclinic structure (insets in Fig. 2c). In the monoclinic structure, the arrangement of Se/S atoms can be viewed as a distorted cubic body-centered lattice. Thus, in the monoclinic–cubic phase transition, the Se/S atoms move slightly by a small external driving force, which is consistent with the negligible thermal hysteresis shown in Fig. 2c. However, the arrangement of Se/S atoms in the orthorhombic structure is quite different from those in either the monoclinic structure or cubic structure. Thus, a higher external driving force is needed, yielding obvious thermal hysteresis (Fig. 2c).

Figure 2d presents a plot of the maximum bending strain ($\varepsilon_{max}$) as a function of S content $x$ with detailed engineering stress–strain curves measured by three-point bending tests shown in Fig. S6. A clear brittle–ductile transition occurs at approximately $x=0.25$–0.30. The thermal history of the material has a negligible effect on this transition. When $x$ is smaller than this threshold, the materials are brittle. However, when $x$ is larger than the threshold, the materials are ductile regardless of whether the structure is orthorhombic or monoclinic. As shown in Figs. S6c, d, the maximum bending and compressive strains of the ductile materials near the MPB (e.g., $Ag_2Se_{0.61}S_{0.39}$ and $Ag_2Se_{0.64}S_{0.36}$) are comparable with those of ductile polycrystalline $Ag_2S$-based materials[1,16,17,19]. These ductile materials in the MPB region can be machined into thin strips and bent and twisted into different complex shapes without cracking (upper right inset of Fig. 2d), showing good flexibility similar to organic materials. The ductility in $Ag_2S$ and the brittleness in $Ag_2Se$ are usually understood from the point of view of chemical bonds[16,18]. It is believed that the Ag-S bonds are multicentered and diffuse, which respond to the easy movement of dislocations and thus good ductility, while the Ag-Se bonds do not have such characteristics. For $Ag_2Se_{1-x}S_x$ pseudobinary compounds, the Ag-S bonds are isolated when $x$ is small, resulting in brittle features. In contrast, when $x$ is large, the Ag-S bonds are percolated, resulting in ductile features (schematics shown in the insets of Fig. 2d). According to the three-dimensional percolation model, the percolation threshold is estimated to be approximately 0.25–0.31[46], which is consistent with the brittle–ductile transition region shown in Fig. 2d.

In contrast to the mechanical properties, the TE properties of $Ag_2Se_{1-x}S_x$ pseudobinary compounds are sensitive to crystal structures instead of the S content. Figs. S7–9 plot the temperature dependences of TE properties for $Ag_2Se_{1-x}S_x$ pseudobinary compounds. Abrupt changes in TE properties are observed in the phase transition range, which originate from the strong electron and phonon scattering caused by critical fluctuations[47,48]. Before the orthorhombic–monoclinic phase transition, $\sigma$ has a weak temperature dependence (Fig. S7), while $\kappa$

increases with increasing temperature, which originates from the increased contribution of electron thermal conductivity ($\kappa_e$) (Fig. S8). Figure 3a–d presents summaries of the room-temperature TE properties of $Ag_2Se_{1-x}S_x$ pseudobinary compounds as a function of S content. The data are listed in Table SII. At room temperature, the $Ag_2Se_{1-x}S_x$ samples crystalizing in the orthorhombic structure have large $\sigma$, $PF$, $\kappa$, and $zT$ values, while those crystalizing in the monoclinic structure have low $\sigma$, $PF$, $\kappa$, and $zT$ values. This phenomenon is more obvious in the MPB region. At the same chemical composition, the $\sigma$ of $Ag_2Se_{1-x}S_x$ crystallized in the orthorhombic structure is 2–3 times larger than that in the monoclinic structure, resulting in higher $PF$ (Fig. 1a) and $zT$ (Fig. 3d) values. Likewise, Fig. S10 shows that the S content $x$ in $Ag_2Se_{1-x}S_x$ has little influence on the carrier concentration ($n$) for the orthorhombic structure but has a large impact on $n$ for the monoclinic structure.

The large divergence in TE properties between orthorhombic and monoclinic structures can be explained by the band edges of the material. The band structures of $Ag_2Se_{0.5}S_{0.5}$ specimens crystallized in orthorhombic and monoclinic structures are shown in Fig. S11a. Although the conduction band minimum (CBM) values of these two structures are mainly contributed by Ag-5s electrons, the CBM of the orthorhombic structure is more disperse than that of the monoclinic structure, leading to the small carrier effective mass ($m^*$) near the value (Table S1 and Fig. 3e). The $m^*$ of $Ag_2Se_{0.5}S_{0.5}$ in the orthorhombic structure is 0.15 m$_e$, which is approximately 1/3 of that in the monoclinic structure. A similar phenomenon can be observed in $Ag_2S$ and $Ag_2Se$ (Fig. 3e and Fig. S11b, c). Although such a small $m^*$ in the orthorhombic structure results in a lower $\alpha$ than that in the monoclinic structure (Fig. S12a), much larger carrier mobility $\mu_H$ values (Fig. 3f and Fig. S12b) are observed, which give rise to the higher $\sigma$, $PF$, and $zT$ values observed in the orthorhombic structure.

The above results indicate that high-performance $Ag_2Se_{1-x}S_x$ pseudobinary ductile TE materials exist near the MPB region with detailed chemical compositions determined by the material's thermal history. Good ductility and high $PF/zT$ can be simultaneously realized when $x$ is 0.25–0.36 and 0.25–0.42 for the heating process and cooling process, respectively. Therefore, by adjusting the material chemical composition and thermal treatment process, the MPB in $Ag_2Se_{1-x}S_x$ pseudobinary compounds can be modulated for the needed mechanical and TE properties. The maximum $PF$ values of ductile orthorhombic $Ag_2Se_{1-x}S_x$ pseudobinary compounds at room temperature reach 22 μWcm⁻¹K⁻², approximately four times that for monoclinic $Ag_2Se_{1-x}S_x$[17]. The maximum $zT$ value at room temperature is 0.61, approximately two times that for monoclinic $Ag_2Se_{1-x}S_x$[17]. These $PF$ and $zT$ values are record-high values (Fig. 1b) compared with the reported organic materials and ductile inorganic TE materials[1,11,17,19–22,29–35,44,49,50]. The maximum $PF$ is comparable to that of $Ag_2Se$-based inorganic–organic hybrid materials[36–38]. In addition, the ductile $Ag_2Se_{1-x}S_x$ materials show good performance stability between 300 K and 400 K and high reproducibility (Figs. S13 and S14).

To confirm the high TE performance in these ductile materials near the MPB region, a flexible in-plane TE device with 6 n/p couples is fabricated taking orthorhombic $Ag_2Se_{0.67}S_{0.33}$ thin strips as n-type legs and Pt–Rh wires as p-type legs (Fig. 4a, b). The thickness and length of these strips are approximately 0.1 mm and 20 mm, respectively. As shown in Fig. 4c, the device has a high material normalized power density of 0.26 Wm⁻¹ under a temperature gradient of 20 K, which is approximately three times that of the $Ag_2Se_{0.5}S_{0.5}$/Pt–Rh device (0.08 Wm⁻¹) and more than two orders of magnitude higher than those of organic TE devices under the same conditions[17,20,51–56] (Fig. 4c). Such a large value is contributed by the very high $PF$ values of orthorhombic TE materials. Likewise, the excellent ductility of orthorhombic $Ag_2Se_{0.67}S_{0.33}$ promises the good flexibility of our device. At a bending radius of 7.5 mm, the internal resistance of the device is scarcely changed after 100 bending cycles (Fig. 4d), indicating high service durability in real-life applications.

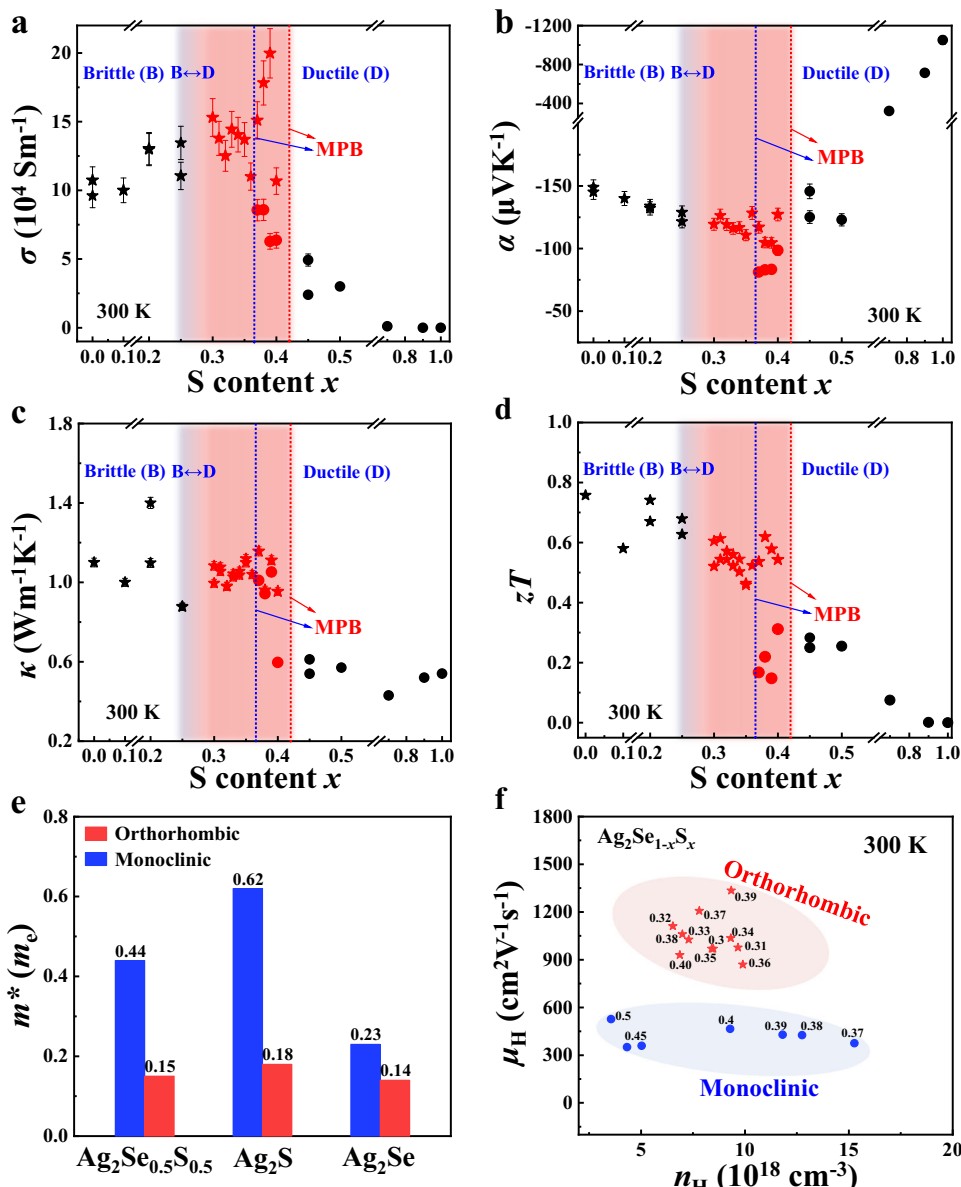

**Fig. 3 | TE properties of Ag₂Se₁₋ₓSₓ pseudobinary compounds.** Room-temperature **a** electrical conductivity ($\sigma$), **b** Seebeck coefficient ($\alpha$), **c** thermal conductivity ($\kappa$), and **d** TE figure of merit ($zT$) as a function of S content. The dashed lines are the MPB in the heating and cooling processes. The error bars in **a**–**c** represent ±9%, ±4%, and ±2%, respectively. **e** Calculated carrier effective mass ($m^*$) near the conduction band minimum for Ag₂Se₀.₅S₀.₅, Ag₂S, and Ag₂Se crystallized in orthorhombic and monoclinic structures. $m^*$ is defined as the arithmetic mean of the carrier effective mass along the Γ-X, Γ-Y, and Γ-Z directions. **f** Room-temperature Hall carrier mobility ($\mu_H$) as a function of the Hall carrier concentration ($n_H$) for orthorhombic and monoclinic Ag₂Se₁₋ₓSₓ.

In summary, we show that the TE performance of ductile materials is pushed to a high value close to that of classic Ag₂Se- and Bi₂Te₃-based materials by modulating the monoclinic–orthorhombic MPBs in Ag₂Se-Ag₂S pseudobinary phases. The flexible TE devices made from such excellent ductile materials demonstrate the outstanding normalized power density and good service durability of the material. The results of this study indicate that ductile TE materials can realize high output power similar to classic brittle TE materials and thus greatly accelerate the application of flexible TE technology for wearable electronics.

## Methods
### Sample preparation
Polycrystalline Ag₂Se₁₋ₓSₓ pseudobinary compounds were prepared from high-purity sliver shots (99.999%, Alfa Aesar), sulfur bulks (99.999%, Alfa Aesar), and selenium shots (99.999%, Alfa Aesar). The stoichiometric mixtures were weighed, sealed in evacuated quartz

tubes under a pressure of 10⁻⁴ Torr, placed in a vertical furnace, heated to 1273 K, and placed at this temperature for 12 h. Then, the tubes were slowly cooled to 723 K with a cooling rate of 20 K h⁻¹, dwelled at 723 K for 12 h, and finally cooled to room temperature. The as-prepared ingots were directly cut into specific shapes. Some of the samples were sealed in evacuated quartz tubes, annealed at 473 K for 48 h, and naturally cooled to room temperature. Some of the samples were dipped in liquid nitrogen (78 K) for 48 h and naturally heated to room temperature. These obtained samples were used for the following characterizations.

### Sample characterization
The element distribution at the microscale was determined by an energy dispersive spectrometer (EDS; ZEISS® Supra 55). As shown in Fig. S15, all elements were homogeneously distributed inside the prepared Ag₂Se₁₋ₓSₓ samples. Furthermore, the EDS mapping

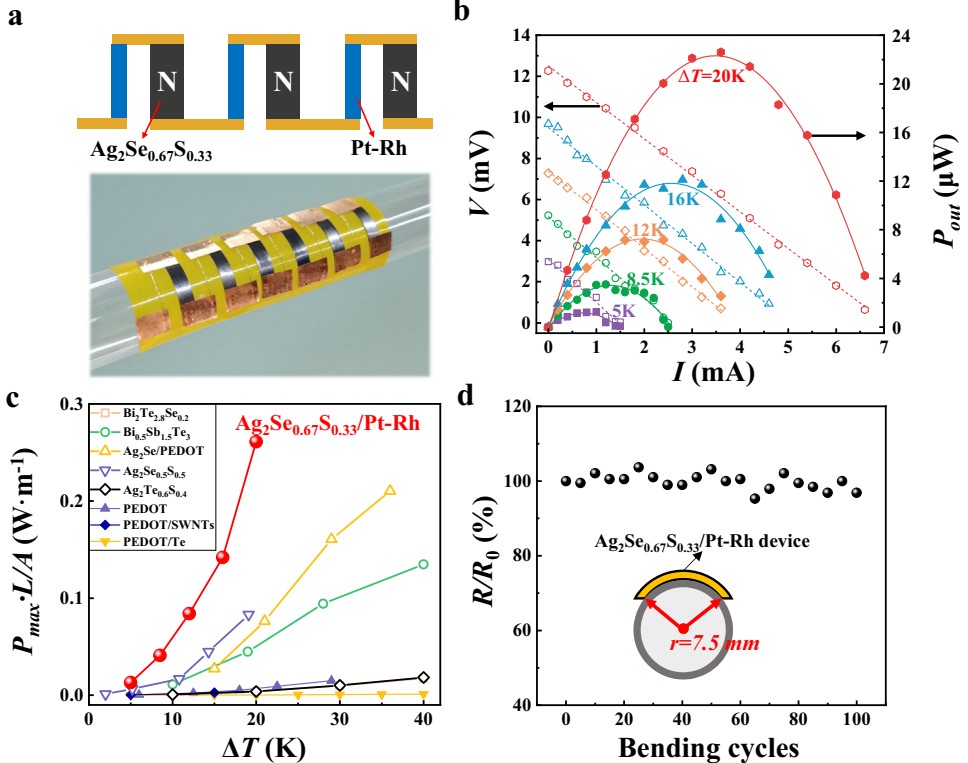

**Fig. 4 | Flexible in-plane Ag$_2$Se$_{0.67}$S$_{0.33}$/Pt−Rh device. a** Sketch map and optical image. **b** Output voltage ($V$) and power output ($P$) as a function of current ($I$) under different operating temperature differences ($\Delta T = 5$ K, 12 K, 16 K, and 20 K). The cold-side temperature is fixed at 296 K. **c** Comparison of the normalized maximum power densities of the materials ($P_{max}L/A$, where $P_{max}$ is the maximum $P$ at each $\Delta T$ and $L$ and $A$ are the length and cross-sectional area of the TE legs, respectively)

among the Ag$_2$Se$_{1-x}$S$_x$-based inorganic TE devices, inorganic−organic hybrid flexible TE devices, and organic flexible TE devices. The data are taken from Refs. 17,20,51−56. **d** Relative electrical resistance variation ($R/R_O$) of the Ag$_2$Se$_{0.67}$S$_{0.33}$/Pt−Rh device after bending for different cycles. The inset shows a sketch map of the bending process. The bending radius is 7.5 mm.

characterized by using TEM equipment (TECNAI F20) indicated that the samples had high homogeneity at the nanoscale (Fig. S16). The heat flow curves were measured by differential scanning calorimetry with a heating/cooling rate of 5 K min⁻¹ (Netzsch® DSC 200F3). As shown in Fig. S17a, the different heating/cooling rates (10 K min⁻¹, 5 K min⁻¹, and 1 K min⁻¹) only influenced the intensities of the endothermic/exothermic peaks but scarcely influenced the initial phase transition temperatures. In addition, Fig. S17b shows that the measured heat flow curves had good repeatability. The phase composition and crystal structure were examined by X-ray diffraction (XRD) analysis (D8 ADVANCE instrument, Bruker Co. Ltd). The electrical conductivity ($\sigma$) and Seebeck coefficient ($\alpha$) were simultaneously measured by modified thermal expansion equipment (Netzsch, DIL 402 C). Details about the thermal conductivity ($\kappa$) measurements can be found elsewhere[17]. The Hall coefficient ($R_H$) was measured by the Van der Pauw method (LakeShore® 8400 series) by sweeping the magnetic field from −0.9 T to +0.9 T at 300 K. The Hall carrier concentration ($n_H$) and Hall mobility ($\mu_H$) were estimated by the relationships $n_H = 1/eR_H$ and $\mu_H = \sigma R_H$, respectively. Bending tests were conducted on a dynamic mechanical analyzer (DMA 850) with a loading rate of 0.05 mm min⁻¹. The specimen dimensions for the bending test were approximately $1.5 \times 0.6 \times 9$ mm³. Compression tests were conducted on an Instrons 5566 universal machine at a loading rate of 0.2 mm min⁻¹. The specimen dimension for the compression test was approximately $3 \times 3 \times 6$ mm³. All specimens were cut directly from the as-prepared ingots.

### Device fabrication and tests
The Ag$_2$Se$_{0.67}$S$_{0.33}$ materials used for the device fabrication were fabricated by the cooling process. Thin films of Ag$_2$Se$_{0.67}$S$_{0.33}$ were

obtained by roller pressing (KJ group, MSK-HRP-1A). The films were sealed in quartz tubes under a pressure of $10^{-4}$ Torr and annealed at 473 K for 48 h. The Ag$_2$Se$_{0.67}$S$_{0.33}$ films were cut into long strips with dimensions of $0.1 \times 3 \times 20$ mm³ by scissors as n-type TE legs. Pt-Rh wires with a diameter of 0.2 mm were used as the p-type legs. Copper foils were used as the electrodes. Spot welding was used to connect copper foil with Pt-Rh wires and Ag$_2$Se$_{0.67}$S$_{0.33}$ strips. The fabrication details of (AgCu)$_{0.995}$Se$_{0.25}$S$_{0.05}$Te$_{0.7}$ could be found elsewhere[1].

The performance of the TE device was measured using a custom-built instrument. By adjusting the resistive load in the circuit, the $I$-$V$ curve was recorded for power output calculations. Under each temperature difference, the maximum power output ($P_{max} = V_{OC}^2/4R_{in}$, wherein $R_{in}$ is the inner resistance of the device) was reached when the resistance of the external electrical load was matched with the internal resistance of the device. A series of temperature differences (5 K, 8.5 K, 12 K, 16 K, and 20 K) was produced by heating one side of the device using a resistance heater, while the cold site was fixed at 296 K.

### Calculations
The first-principles calculations were performed based on density functional theory (DFT) as implemented in the VASP program[57]. Projector augmented wave (PAW)[58] potentials were used to describe core−valence interactions, and plane waves reaching a kinetic energy of 350 eV were used as the basis set. The PBE functional[59] was used to optimize the atomic structures and calculate the energy with the DFT-D3 method utilized to describe the van der Waals interactions[60]. The atomic structures of Ag$_2$Se$_{1-x}$S$_x$ in both monoclinic and orthorhombic phases were constructed based on the primitive cells of Ag$_2$S (monoclinic) and Ag$_2$Se (orthorhombic). The cell and atomic structures were relaxed until the residual forces on all atoms were smaller than 0.01 eV,

and the energy was calculated. The modified Becke–Johnson exchange potential was used in the band structure calculation[61,62] to improve the accuracy of band gap estimation. A $6 \times 4 \times 4$ k-point mesh was used to relax the atomic structure, and a $10 \times 6 \times 6$ k-point mesh was used to calculate the charge density. The effective masses were calculated by fitting the electronic band structure to quadratic functions near the CBM (both are at the $\Gamma$ point) along the three principal axes.

## Data availability

All data are available in the manuscript and in the supplementary materials.

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

## Acknowledgements

This work was supported by the National Key Research and Development Program of China (2023YFB3809400, T.Z.) and the National Natural Science Foundation of China (grants 52122213, P.Q., 91963208, L.C., and 52232010, X.S.). X.S. is thankful for the support from the Shanghai Pilot Program for Basic Research-Chinese Academy of Science Shanghai Branch (JCYJ-SHFY-2022-002, X.S.) and the Shanghai Government (20JC1415100, X.S.).

## Author contributions

P.Q. and X.S. designed the study. J.S. L and J.L. prepared the samples and measured the thermoelectric and mechanical properties. J.L. fabricated the devices and measured the output performance. C.M. performed the first-principles calculations. J.S.L., J.L., Z.Z. Z.G. and K.Z. collected the data and provided explanations under the guidance of P.Q., L.C., and X.S. J.S.L., J.L., P.Q., K.Z., L.C., and X.S. wrote and revised the paper. All authors discussed the results and provided helpful suggestions for this work.

## Competing interests

The authors declare no competing interests.
