## [Peer Review File · Nature Communications]

Modulation of Morphotropic Phase Boundary for high-performance ductile thermoelectric materialsREVIEWER COMMENTS

Reviewer #1 (Remarks to the Author):

In this work, Liang et al realized an excellent room-temperature ZT value of 0.61 in Ag₂Se-Ag₂S pseudo-binary compounds by tuning the Morphotropic Phase Boundary (MPB). Such a high value can attribute to the ultra-high PF of 22 $\mu\text{W cm}^{-1} \text{K}^{-2}$, which is comparable with that of Ag₂Se-based thermoelectric materials. Furthermore, the orthorhombic Ag₂Se_{1-x}S_x compounds near the MPB possess good ductility, making them promising candidates for flexible thermoelectric materials. The manuscript is well-written and the results are also very interesting. I recommend authors complete major revisions prior to publication:

1. In Figure. 2(b). The XRD result indicates that Ag₂Se_{0.61}S_{0.39} compound is not in pure phase at 300 K and 350 K. Has the Rietveld refinement been followed?
2. The structural data are strongly recommended in this manuscript, such as the XRD patterns of samples with different S contents. Besides, the current elemental distribution data of Ag₂Se_{1-x}S_x compounds in Figure. S8 are not convincing enough, the high-resolution EDS mapping and elemental distribution are needed in the revision.
3. The discussions about thermoelectric performance are not comprehensive. The temperature-dependent thermal conductivity (κ_e and κ_l) of samples with different S contents ($0.3 < x < 0.4$) between 200 K and 375 K should be included.
4. S-alloying can inevitably lead to the reduction of carrier concentration, therefore influencing the electrical transport performance. Please add the temperature-dependent carrier concentration of samples with different S contents ($0.3 < x < 0.4$) between 200 K and 375 K.
5. The current presentation of thermoelectric performance is not clear and hard to read. I suggest the authors add a table of thermoelectric performance at room temperature, especially for samples near the MPB.
6. The authors claimed the maximum PF of 22 $\mu\text{W cm}^{-1} \text{K}^{-2}$ can be achieved for orthorhombic Ag₂Se_{1-x}S_x compounds, which is about four times for monoclinic Ag₂Se_{1-x}S_x. To our best knowledge, Ag₂Se_{1-x}S_x compounds are difficult to have performance stability. The authors should provide proof of the performance stability of the Ag₂Se_{1-x}S_x material in the paper. How is the performance stability of this material? Has the cyclic measurement of thermoelectric performance been conducted? How about the reproducibility of the sample possessing the highest ZT value?

Reviewer #2 (Remarks to the Author):

The manuscript details the results of a study on Ag₂Se-Ag₂S pseudo binary compounds with enhanced power factors (PF) and zT figure of merit which are claimed to be superior to all current flexible TE materials. A thorough analysis of the literature reveals that this claim cannot be upheld if the values for the PF and zT are compared against other studies on similar materials e.g. flexible Ag₂Se films were reported with a PF of 2450.9 $\mu\text{W cm}^{-1} \text{K}^{-2}$. Similarly much higher PFs were obtained for flexible inorganic-organic hybrid materials (e.g. Te nanowires in PEDOT-PSS). In addition it is not clear how reproducible the results are reported in the present study as error bars are missing from all plots which calls into question their scientific validity. The manuscript is also poorly written with numerous grammatical errors and use of incorrect English terms. I therefore cannot recommend publication of this manuscript and recommend that the authors thoroughly revise their paper and in addition perform more measurements to verify the reproducibility of their results.

Reviewer #3 (Remarks to the Author):

Here, the authors demonstrated that Morphotropic Phase Boundary (MPB) lies in Ag₂Se-Ag₂S pseudo-binary compounds alike that in ferroelectrics, which can be freely tuned through adjusting thermal treatment processes. High-performance ductile TE materials with excellent PF ($22 \mu\text{Wcm}^{-1}\text{K}^{-2}$) and zT (0.61) values are realized at 300 K near the MPB, superior to all current flexible TE materials.

After carefully reading through this manuscript, I have found it comprehensive and worth publication after some minor revisions addressing the following comments:

1. Is it possible the difference between heating/cooling due to the nature of measurement process, i.e. the slag of temperature change within the material when the cooling/heating rate is too fast?
2. The heating/cooling patterns need to be repeatedly measured to assured the repeatability.
3. Can MPB be directly observed and demonstrated by TEM? If yes, why not putting some direct evidence here, this might be helpful.
4. As MPB is the key concept here, it is necessary to highlight the uniqueness of MPB and its relationship with the high performance achieved here. Simply addressing a rarely discussed concept is not scientific enough.

Reply to the First Reviewer's report:

1. In Figure. 2(b). The XRD result indicates that $\text{Ag}_2\text{Se}_{0.61}\text{S}_{0.39}$ compound is not in pure phase at 300 K and 350 K. Has the Rietveld refinement been followed?

Response: We have performed the Rietveld refinement on the XRD patterns of $\text{Ag}_2\text{Se}_{0.61}\text{S}_{0.39}$ collected at 300 K and 350 K. The results are shown in Fig. R1 and the residual parameters are listed in Table R1. For the XRD pattern collected at 300 K, all the diffraction peaks can be indexed with the orthorhombic structure ($P2_12_12_1$). The full profile refinement converges with the respectable values of the residuals $R_p = 8.91\%$ and $wR_p = 11.44\%$. For the XRD pattern collected at 350 K, all the diffraction peaks can be indexed with the monoclinic structure ($P2_1/n$). The full profile refinement converges with the respectable values of the residuals $R_p = 5.96\%$ and $R_p = 7.60\%$. These results prove that phases of $\text{Ag}_2\text{Se}_{0.61}\text{S}_{0.39}$ at 300 K and 350 K are both pure.

In the resubmitted manuscript, Fig. R1, Table R1, and the related discussions were added.

Fig. R1 Powder X-ray diffraction patterns of $\text{Ag}_2\text{Se}_{0.61}\text{S}_{0.39}$ collected at (a) 300 K and (b) 350 K. The black circles represent the measured point intensities. The red lines represent the intensities calculated from the assigned structures. The blue lines show the difference between experimental and calculated intensities. The black vertical bars indicate Bragg positions.

Table R1. Refined residual parameters for $\text{Ag}_2\text{Se}_{0.61}\text{S}_{0.39}$ at 300 K and 350 K.

T (K)	Space group	$R(\text{obs})$	$wR(\text{obs})$	$R(\text{all})$	$wR(\text{all})$	R_p	wR_p
300	$P2_12_12_1$	3.88%	4.22%	3.95%	4.23%	8.91%	11.44%
350	$P2_1/n$	4.97%	5.45%	5.98%	5.77%	5.96%	7.60%

2. *The structural data are strongly recommended in this manuscript, such as the XRD patterns of samples with different S contents. Besides, the current elemental distribution data of $\text{Ag}_2\text{Se}_{1-x}\text{S}_x$ compounds in Figure. S8 are not convincing enough, the high-resolution EDS mapping and elemental distribution are needed in the revision.*

Response: Thanks for your kind suggestion. Fig. R2a shows the XRD patterns of $\text{Ag}_2\text{Se}_{1-x}\text{S}_x$ pseudo-binary compounds with different S contents. When the S content $x < 0.2$, all diffraction peaks can be well indexed belonging to the orthorhombic structure of Ag_2Se . When $x \geq 0.4$, all diffraction peaks can be well indexed belonging to the monoclinic structure of Ag_2S . When x is near the morphotropic phase boundary, such as $x = 0.2, 0.25, 0.3, \text{ and } 0.35$, the diffraction peaks belonging to the orthorhombic structure and monoclinic structure are simultaneously observed. These results are consistent with the Ag_2Se - Ag_2S phase diagram determined in this work. Likewise, the enlarged XRD patterns at $2\theta = 30^\circ\text{-}35^\circ$ show that the diffraction peaks gradually shift to the right with increasing the S content, indicating the shrinkage of the lattice. This is reasonable since S has smaller atomic radius than Se.

As suggested by the reviewer, we also measured the high-resolution EDS mapping of $\text{Ag}_2\text{Se}_{1-x}\text{S}_x$ pseudo-binary compounds. As examples, Fig. R3 shows the EDS mapping performed on $\text{Ag}_2\text{Se}_{1-x}\text{S}_x$ ($x = 0.30, 0.35, 0.39, \text{ and } 0.5$) under the magnification $\times 10000$ by using the SEM equipment. All elements are also

homogeneously distributed in the microscale. No element enrichment can be observed, even for the $\text{Ag}_2\text{Se}_{0.65}\text{S}_{0.35}$ and $\text{Ag}_2\text{Se}_{0.61}\text{S}_{0.39}$ with the composition near the morphotropic phase boundary. Fig. R4 further shows the EDS mapping of $\text{Ag}_2\text{Se}_{0.7}\text{S}_{0.3}$ characterized by using the TEM equipment. All elements are also homogeneously distributed in the nanoscale. Thus, it can be concluded that our $\text{Ag}_2\text{Se}_{1-x}\text{S}_x$ samples have high homogeneity.

In the resubmitted manuscript, Fig. R2 and the related discussions were added. The initial low-resolution EDS mapping was replaced by the high-resolution EDS mapping shown in Figs. R3-4.

Fig. R2 Room-temperature XRD patterns of as-prepared bulk $\text{Ag}_2\text{Se}_{1-x}\text{S}_x$ pseudo-binary compounds. The right panel shows the magnification at $2\theta = 30^\circ\text{-}35^\circ$.

Fig. R3 Backscatter electron (AsB) image and elemental energy dispersive spectroscopy (EDS) mappings of $\text{Ag}_2\text{Se}_{1-x}\text{S}_x$ ($x = 0.30, 0.35, 0.39,$ and 0.5) characterized by SEM equipment.

Fig. R4 Elemental energy dispersive spectroscopy (EDS) mappings of $\text{Ag}_2\text{Se}_{0.7}\text{S}_{0.3}$ characterized by TEM equipment. The specimen was fabricated by using the ultrathin

section technique. Due to the material's intrinsic ductility, the specimen is not very flat, which is responsible for the contrast shown in the figures.

3. The discussions about thermoelectric performance are not comprehensive. The temperature-dependent thermal conductivity (κ_e and κ_L) of samples with different S contents ($0.3 < x < 0.4$) between 200 K and 375 K should be included.

Response: Thanks for your kind suggestion. As suggested by the reviewer, the total thermal conductivity (κ), electron thermal conductivity (κ_e), and lattice thermal conductivity (κ_L) of $\text{Ag}_2\text{Se}_{1-x}\text{S}_x$ ($0.3 < x < 0.4$) between 200 K and 375 K were plotted in Fig. R5. The κ , κ_e , and κ_L curves show abrupt change during the orthorhombic-monoclinic-cubic phase transitions. Similar phenomenon has been also observed in Cu_2Se (*Adv. Mater.* 2013, 25, 6607; *Adv. Mater.* 2019, 31, 1806518). The strong electron and phonon scattering caused by the fluctuations in the sample density and crystal structure during the phase transition are responsible for these abrupt changes. Before the phase transition, the κ of all samples increases with increasing temperature, which is originated from the increased contribution of carriers to thermal transports. The κ_L shows weak temperature dependence, which is a common phenomenon for Ag- and Cu-based liquid-like thermoelectric materials.

In the resubmitted manuscript, Fig. R5 and the related discussions were added.

Fig. R5 Temperature dependences of (a) thermal conductivity (κ), (b) electron lattice thermal conductivity (κ_e), and (c) lattice thermal conductivity (κ_L) of $\text{Ag}_2\text{Se}_{1-x}\text{S}_x$ ($0.3 < x < 0.4$).

4. S -alloying can inevitably lead to the reduction of carrier concentration, therefore

influencing the electrical transport performance. Please add the temperature-dependent carrier concentration of samples with different S contents ($0.3 < x < 0.4$) between 200 K and 375 K.

Response: Thanks for your kind suggestion. Fig. R6a shows the temperature-dependent carrier concentration (n) of $\text{Ag}_2\text{Se}_{1-x}\text{S}_x$ ($0.3 < x < 0.4$). The n roughly increases with increasing temperature, showing the typical semiconducting feature. During the orthorhombic-monoclinic-cubic phase transitions, slight fluctuations are observed. In addition, we plot the relationship between room-temperature n and S-alloying content x for $\text{Ag}_2\text{Se}_{1-x}\text{S}_x$ ($0 \leq x \leq 1$). As shown in Fig. R6b, the effect of S-alloying is related to the crystal structure. For the $\text{Ag}_2\text{Se}_{1-x}\text{S}_x$ samples crystallizing in the monoclinic structure, S-alloying leads to the reduction of n , which is consistent with that mentioned by the reviewer. However, for the $\text{Ag}_2\text{Se}_{1-x}\text{S}_x$ samples crystallizing in the orthorhombic structure, the n has a weak dependence with x . The investigation on this abnormal phenomenon is underway in our group, but we think they are beyond the scope of the present work.

In the resubmitted manuscript, Fig. R6 and the related discussions were added.

Fig. R6 (a) Temperature dependence of carrier concentration (n) of $\text{Ag}_2\text{Se}_{1-x}\text{S}_x$ ($0.3 < x < 0.4$). (b) Room-temperature n as a function of S-alloying content x for $\text{Ag}_2\text{Se}_{1-x}\text{S}_x$.

5. The current presentation of thermoelectric performance is not clear and hard to read. I suggest the authors add a table of thermoelectric performance at room temperature, especially for samples near the MPB.

Response: Thanks for your kind suggestion. The room-temperature thermoelectric

properties of $\text{Ag}_2\text{Se}_{1-x}\text{S}_x$ are summarized in Table R2. This table was also added in the resubmitted manuscript.

Table R1 Room-temperature thermoelectric properties of $\text{Ag}_2\text{Se}_{1-x}\text{S}_x$ samples with the composition near the MPB.

Composition	S ($\mu\text{V}\cdot\text{K}^{-1}$)	σ ($\text{S}\cdot\text{m}^{-1}$)	κ	κ_e	κ_L	PF ($\mu\text{W}\cdot\text{cm}^{-1}\cdot\text{K}^{-2}$)	zT
$\text{Ag}_2\text{Se}_{0.6}\text{S}_{0.4}$	-99	6.4×10^4	0.60	0.37	0.23	6.2	0.31
$\text{Ag}_2\text{Se}_{0.61}\text{S}_{0.39}$	-83	6.3×10^4	1.05	0.37	0.68	4.4	0.15
$\text{Ag}_2\text{Se}_{0.62}\text{S}_{0.38}$	-83	8.6×10^4	0.94	0.51	0.43	5.9	0.22
$\text{Ag}_2\text{Se}_{0.63}\text{S}_{0.37}$	-81	8.6×10^4	1.01	0.51	0.50	5.6	0.17
$\text{Ag}_2\text{Se}_{0.64}\text{S}_{0.36}$	-128	1.1×10^5	1.04	0.60	0.44	18.1	0.52
$\text{Ag}_2\text{Se}_{0.65}\text{S}_{0.35}$	-111	1.4×10^5	1.10	0.77	0.32	16.8	0.46
$\text{Ag}_2\text{Se}_{0.66}\text{S}_{0.34}$	-117	1.4×10^5	1.06	0.79	0.27	19.2	0.54
$\text{Ag}_2\text{Se}_{0.67}\text{S}_{0.33}$	-116	1.4×10^5	1.04	0.81	0.23	19.5	0.56
$\text{Ag}_2\text{Se}_{0.68}\text{S}_{0.32}$	-119	1.3×10^5	0.98	0.70	0.28	17.8	0.54
$\text{Ag}_2\text{Se}_{0.69}\text{S}_{0.31}$	-126	1.4×10^5	1.08	0.76	0.32	22.0	0.61
$\text{Ag}_2\text{Se}_{0.7}\text{S}_{0.3}$	-120	1.5×10^5	1.08	0.85	0.23	21.8	0.60

6. The authors claimed the maximum PF of $22 \mu\text{W cm}^{-1} \text{K}^{-2}$ can be achieved for orthorhombic $\text{Ag}_2\text{Se}_{1-x}\text{S}_x$ compounds, which is about four times for monoclinic $\text{Ag}_2\text{Se}_{1-x}\text{S}_x$. To our best knowledge, $\text{Ag}_2\text{Se}_{1-x}\text{S}_x$ compounds are difficult to have performance stability. The authors should provide proof of the performance stability of the $\text{Ag}_2\text{Se}_{1-x}\text{S}_x$ material in the paper. How is the performance stability of this material? Has the cyclic measurement of thermoelectric performance been conducted? How about the reproducibility of the sample possessing the highest zT value?

Response: Thanks for your kind suggestion. Since the main application of $\text{Ag}_2\text{Se}_{1-x}\text{S}_x$

samples is wearable electronics, the working temperature range is roughly below 370 K (less than 100 °C). The performance stability of the present $\text{Ag}_2\text{Se}_{1-x}\text{S}_x$ samples are very good below 370 K. As an example, we performed the cyclic measurement of electrical transport properties between 300 K and 370 K on $\text{Ag}_2\text{Se}_{0.69}\text{S}_{0.31}$, which has the highest room-temperature zT among the $\text{Ag}_2\text{Se}_{1-x}\text{S}_x$ samples. As shown in Figs. R7a-b, the thermoelectric properties of the three times measurements are quite similar with each other. Similar good performance stability has been also obtained for $\text{Ag}_2\text{Se}_{0.65}\text{S}_{0.35}$ (Figs. R7c-d).

We have also repeated the preparation of these samples with different compositions at least three times with the same fabrication methods described in the manuscript. The measured TE properties are reproducible. As examples, Fig. R8 shows the thermoelectric properties of three batches of $\text{Ag}_2\text{Se}_{0.69}\text{S}_{0.31}$. The differences among different batches are very small.

In the resubmitted manuscript, Figs. R7-8 and the related discussions were added.

Fig. R7 Cyclic measurements on electrical transport properties of (a) $\text{Ag}_2\text{Se}_{0.69}\text{S}_{0.31}$ and (b) $\text{Ag}_2\text{Se}_{0.65}\text{S}_{0.35}$ between 300 K and 370 K.

Fig. R8 Thermoelectric properties of three batches of $\text{Ag}_2\text{Se}_{0.69}\text{S}_{0.31}$ samples.

Reply to the Second Reviewer's report:

1. The manuscript details the results of a study on $\text{Ag}_2\text{Se}-\text{Ag}_2\text{S}$ pseudo binary compounds with enhanced power factors (PF) and zT figure of merit which are claimed to be superior to all current flexible TE materials. A thorough analysis of the literature reveals that this claim cannot be upheld if the values for the PF and zT are compared against other studies on similar materials e.g. flexible Ag_2Se films were reported with a PF of 2450.9 $\mu\text{Watts}/\text{mK}^2$. Similarly, much higher PFs were obtained for flexible inorganic-organic hybrid materials (e.g. Te nanowires in PEDOT-PSS).

Response: We are sorry that we did not clarify our comparison objects clearly in the

manuscript. In this work, we intend to compare the PF and zT of ductile $Ag_2Se_{1-x}S_x$ samples with the reported ductile inorganic semiconductors and organic materials, rather than all flexible materials. As shown in Fig. R9, the PF ($\sim 20 \mu Wcm^{-1}K^{-2}$) and zT (~ 0.6) are indeed the highest values among these materials. In the resubmitted manuscript, the statement that ‘superior to all current flexible TE materials’ was corrected into ‘superior to all current ductile inorganic semiconductors and organic materials’.

As suggested by the reviewer, we have carefully re-surveyed the literatures about the flexible inorganic-organic hybrid thermoelectric materials. We found the reported PF values of all Te/PEDOT-PSS composites are less than $3 \mu Wcm^{-1}K^{-2}$ (or $300 \mu Wm^{-1}K^{-2}$, using the unit mentioned by the reviewer), much lower than those of the $Ag_2Se_{1-x}S_x$ samples reported in this work. Recently, very high PF values were indeed reported for the flexible Ag_2Se films, such as $24.5 \mu Wcm^{-1}K^{-2}$ (or $2450.9 \mu Wm^{-1}K^{-2}$) reported by Gao et al. on *iScience* (2020) and $22.3 \mu Wcm^{-1}K^{-2}$ (or $2230 \mu Wm^{-1}K^{-2}$) reported by Cai et al. on *Energy Environ. Sci.* (2020). However, Ag_2Se is intrinsically brittle. The flexibility of these Ag_2Se -based films relies on the flexible organic substrate (e.g. paper and nylon) and the thin thickness of Ag_2Se film (just about several micrometers). Thus, these data were not included in Fig. R9, which only includes the data of ductile inorganic semiconductors and organic materials.

In the resubmitted manuscript, several sentences were added to discuss the difference between the flexible Ag_2Se -based films and the ductile $Ag_2Se_{1-x}S_x$ samples investigated in this work. The related literatures, including those mentioned by the reviewer, were also added in the Reference list.

Fig. R9 Room-temperature TE figure of merit (zT) as a function of power factor (PF) for ductile orthorhombic $\text{Ag}_2\text{Se}_{1-x}\text{S}_x$ ($0.30 \leq x \leq 0.35$) pseudo-binary compounds. The data of other typical ductile organic semiconductors and organic materials reported before are included for comparison.

2. In addition it is not clear how reproducible the results are reported in the present study as error bars are missing from all plots which calls into question their scientific validity.

Response: Please see the answer to the Question 6 asked by the first reviewer. We have repeated the preparation of the $\text{Ag}_2\text{Se}_{1-x}\text{S}_x$ samples with different compositions for three times. The measured TE properties are reproducible. As an example, Fig. R8 shows the thermoelectric properties of three batches of $\text{Ag}_2\text{Se}_{0.69}\text{S}_{0.31}$ samples. The differences among different batches are very small. At 300 K, the maximum deviations are about $\pm 4\%$, $\pm 9\%$, and $\pm 2\%$ for Seebeck coefficient, electrical conductivity, and thermal conductivity, respectively. Likewise, we also measured the thermoelectric performance of one $\text{Ag}_2\text{Se}_{0.69}\text{S}_{0.31}$ sample for three times. As shown in Fig. R7a, the thermoelectric properties of the three times measurements are quite similar with each other. The maximum deviations of Seebeck coefficient and electrical conductivity are $\pm 4\%$ and ± 3

at 300 K. Similar good repeatability has been also obtained for $\text{Ag}_2\text{Se}_{0.65}\text{S}_{0.35}$ (Fig. R7b). These results prove that our measured results in this work are reproducible and repeatable.

In the resubmitted manuscript, the error bars were added in Figs. 3a-c according to the maximum deviations of Seebeck coefficient, electrical conductivity, and thermal conductivity obtained in the three batches of $\text{Ag}_2\text{Se}_{0.69}\text{S}_{0.31}$ samples.

3. The manuscript is also poorly written with numerous grammatical errors and use of incorrect English terms.

Response: Sorry for our carelessness. We have thoroughly revised our manuscript and corrected the grammatical errors and improper English terms, as you can find in the resubmitted manuscript.

Reply to the Third Reviewer's report:

1. Is it possible the difference between heating/cooling due to the nature of measurement process, i.e. the slag of temperature change within the material when the cooling/heating rate is too fast?

Response: This is a good question. In this work, the Ag_2Se - Ag_2S phase diagram was obtained by measuring the heat flow curves with the heating/cooling rate of 5 Kmin^{-1} . Taking $\text{Ag}_2\text{Se}_{0.69}\text{S}_{0.31}$ as an example, we have also tried other heating/cooling rates such as 10 Kmin^{-1} and 1 Kmin^{-1} to measure the heat flow curves. As shown in Fig. R10, the different heating/cooling rates only influence the intensity of the endothermic/exothermic peaks, but scarcely influence the initial phase transition temperatures. This indicates that the difference in the phase diagram between heating/cooling processes is originated from the material's intrinsic property rather than the measurement. In the resubmitted manuscript, Fig. R10 and the related discussions were added.

Fig. R10 Heat flow curves of $\text{Ag}_2\text{Se}_{0.69}\text{S}_{0.31}$ during heating and cooling processes. The heating/cooling rates are 1 Kmin^{-1} , 5 Kmin^{-1} , and 10 Kmin^{-1} , respectively.

2. *The heating/cooling patterns need to be repeatedly measured to assured the repeatability.*

Response: Thank you for your kind suggestion. We have repeatedly measured the heat flow curves of the $\text{Ag}_2\text{Se}_{1-x}\text{S}_x$ samples with different compositions. The heating/cooling patterns are repeatable. As an example, Fig. R11 shows the heat flow curves of $\text{Ag}_2\text{Se}_{0.69}\text{S}_{0.31}$ with the heating/cooling rate of 5 Kmin^{-1} . The differences among three times measurement are very small. In the resubmitted manuscript, Fig. R11 and the related discussions were added.

Fig. R11 Cyclic measurements on the heat flow curves of $\text{Ag}_2\text{Se}_{0.69}\text{S}_{0.31}$.

3. *Can MPB be directly observed and demonstrated by TEM? If yes, why not putting some direct evidence here, this might be helpful.*

Response: We have tried to use the TEM to directly characterize the MPB. Unfortunately, the monoclinic structure and orthorhombic structure are difficult to be distinguished by TEM. As shown in Fig. R12, the selected area electron diffraction (SAED) pattern obtained from the fast Fourier transformation (FFT) can be well identified as either monoclinic structure or orthorhombic structure. Thus, we did not put the TEM results in this work. In the resubmitted manuscript, one sentence was added addressing this issue.

Fig. R12 High resolution transmission electron microscopy (TEM) image of $\text{Ag}_2\text{Se}_{0.69}\text{S}_{0.31}$. The upper left panel shows the selected area electron diffraction pattern obtained by fast Fourier transformation.

4. As MPB is the key concept here, it is necessary to highlight the uniqueness of MPB and its relationship with the high performance achieved here. Simply addressing a rarely discussed concept is not scientific enough.

Response: We are sorry that we did not display the uniqueness of MPB and its relationship with the high performance clearly. Morphotropic Phase Boundary (MPB) is a transition region in the oblique temperature-composition phase diagram which separates two competing phases with different crystallographic symmetries. At elevated temperature, both these two competing phases can convert into the same phase with another crystallographic symmetry. The MPB observed in Ag_2Se - Ag_2S phase diagram is very unique since it has been never observed in other thermoelectric materials before.

Due to the comparable energies of the two competing phases, abnormal physical properties were often observed near the MPB. In this work, good ductility was observed in the orthorhombic $\text{Ag}_2\text{Se}_{1-x}\text{S}_x$ samples near the MPB. This is quite unexpected since the orthorhombic $\text{Ag}_2\text{Se}_{1-x}\text{S}_x$ samples are believed to be brittle for a long time. Likewise, the small carrier effective mass near the conduction band minimum in orthorhombic $\text{Ag}_2\text{Se}_{1-x}\text{S}_x$ gives rise to the large carrier mobility, yielding high σ , PF , and zT . Thus, the orthorhombic $\text{Ag}_2\text{Se}_{1-x}\text{S}_x$ pseudo-binary compounds near the MPB can well integrate the excellent thermoelectric performance and good ductility. In the resubmitted manuscript, these points were highlighted.

List of changes

1. The expression of “flexible TE materials” was corrected into “ductile inorganic semiconductors and organic materials”.
2. Rietveld refinements were performed on the powder X-ray diffraction patterns of

$\text{Ag}_2\text{Se}_{0.61}\text{S}_{0.39}$ collected at different temperatures. The results and related discussions were added.

3. The X-ray diffraction patterns performed on the bulk $\text{Ag}_2\text{Se}_{1-x}\text{S}_x$ were added.
4. The error bars of TE properties were added in Fig. 3.
5. The temperature dependences of thermal conductivity and carrier concentration for $\text{Ag}_2\text{Se}_{1-x}\text{S}_x$ pseudo-binary compounds and the related discussions were added.
6. The relationship between S content and carrier concentration in $\text{Ag}_2\text{Se}_{1-x}\text{S}_x$ ($0.3 < x < 0.4$) was added.
7. The difference between the Ag_2Se -based inorganic-organic hybrid materials and present ductile $\text{Ag}_2\text{Se}_{1-x}\text{S}_x$ was discussed.
8. The performance stability and reproducibility of $\text{Ag}_2\text{Se}_{1-x}\text{S}_x$ were measured and discussed.
9. High resolution EDS mapping results were added.
10. The heat flow curves with different heating/cooling rates were added.
11. The repeatability of the DSC measurement was added.
12. Several new references were added.

REVIEWER COMMENTS

Reviewer #1 (Remarks to the Author):

All the reviewers' comments were clearly addressed in the revised manuscript. Therefore I recommend that this manuscript is proper to be published on NC.

Reviewer #2 (Remarks to the Author):

Even though the authors state that the manuscript has been thoroughly revised to correct numerous English grammatical errors and typos I still found that the current version is still not very readable when it comes to the use of English.

I believe that the PFs for Ag₂Se films, such as 24.5 $\mu\text{Wcm}^{-1}\text{K}^{-2}$ (or 2450.9 $\mu\text{Wm}^{-1}\text{K}^{-2}$) reported by Gao et al. on iScience (2020) and 22.3 $\mu\text{Wcm}^{-1}\text{K}^{-2}$ (or 2230 $\mu\text{Wm}^{-1}\text{K}^{-2}$) reported by Cai et al. on Energy Environ. Sci. (2020) need to be included in Figure R9 to make a fair comparison with other flexible thermoelectric materials.

As it stands the manuscript is still not publishable and would need further revision

Reviewer #3 (Remarks to the Author):

All the raised questions have been addressed. I suggest accepting this paper for publication.

Reply to the Second Reviewer's report:

1. *Even though the authors state that the manuscript has been thoroughly revised to correct numerous English grammatical errors and typos I still found that the current version is still not very readable when it comes to the use of English.*

Response: Thanks for your kind suggestion. We have carefully revised our manuscript to improve the readability in the resubmitted manuscript.

2. *I believe that the PFs for Ag₂Se films, such as 24.5 $\mu\text{Wcm}^{-1}\text{K}^{-2}$ (or 2450.9 $\mu\text{Wm}^{-1}\text{K}^{-2}$) reported by Gao *et al.* on *iScience* (2020) and 22.3 $\mu\text{Wcm}^{-1}\text{K}^{-2}$ (or 2230 $\mu\text{Wm}^{-1}\text{K}^{-2}$) reported by Cai *et al.* on *Energy Environ. Sci.* (2020) need to be included in Figure R9 to make a fair comparison with other flexible thermoelectric materials.*

Response: Thanks for your kind suggestion. As shown in Fig. R1, the PFs and zTs of the Ag₂Se-based films reported by Gao *et al.* and Cai *et al.* were added. It should be noted that these two works did not report the accurate zTs . The zTs in Fig. R1 were estimated by using the in-plane thermal conductivity provided by the authors. In the resubmitted manuscript, the Fig. 1a was updated by Fig. R1.

Fig. R1 Room-temperature TE figure of merit (zT) as a function of power factor (PF) for ductile orthorhombic $\text{Ag}_2\text{Se}_{1-x}\text{S}_x$ ($0.30 \leq x \leq 0.35$) pseudo-binary compounds. The data of other typical ductile organic semiconductors, organic materials, and Ag_2Se -based films reported before are included for comparison.

List of changes

1. The English writing was improved to improve the readability.
2. In the resubmitted manuscript, the Fig. 1a was updated by Fig. R1.

REVIEWER COMMENTS

Reviewer #2 (Remarks to the Author):

It is good to see that the authors have taken the comments onboard and have now included the zT and PF values for Ag₂Se films in Fig. 1. However in the paper by Cai et al. published in Energy Env. Sci. a ZT of 0.5 has been quoted for Cu₁Ag₄Se₃ film on a nylon membrane which is higher than what is shown in Fig.1. This needs to be corrected.

In addition there are still multiple grammatical errors present despite assurances that the manuscript has been thoroughly proof-read. This is unacceptable after two rounds of revision and I cannot recommend publication for this reason.

Reply to the Second Reviewer's report:

1. It is good to see that the authors have taken the comments onboard and have now included the zT and PF values for Ag_2Se films in Fig. 1. However, in the paper by Cai et al. published in *Energy Environ. Sci.* a ZT of 0.5 has been quoted for $Cu_1Ag_4Se_3$ film on a nylon membrane which is higher than what is shown in Fig.1. This needs to be corrected.

Response: Thanks for pointing out this. In 2020, Cai et al. reported a n-type $Ag_2Se/Ag/CuAgSe$ thermoelectric (TE) composite film supported by a porous nylon membrane with a power factor $2231.5 \mu W m^{-1} K^{-2}$ and $ZT \sim 0.5$ at 300 K (*Energy Environ. Sci.*, 2020, 13, 1240). Later, they published a Correction on these results (*Energy Environ. Sci.*, 2020, 13, 1287, please see Fig. R1). They claimed that the power factor '2231.5 $\mu W m^{-1} K^{-2}$ ' should read as '1593.9 $\mu W m^{-1} K^{-2}$ ' and the ' $ZT \sim 0.5$ ' should read as ' $ZT \sim 0.4$ '. Please see the following figures. Therefore, in our manuscript, we used the corrected PF and ZT values to plot the figure. In the resubmitted manuscript, both these two references from Cai et al. were cited.

Energy &
Environmental
Science

CORRECTION

View Article Online
View Journal | View Issue

Cite this: *Energy Environ. Sci.*,
2020, 13, 1287

DOI: 10.1039/d0ee90012e

rsc.li/ees

Correction: Ultrahigh power factor and flexible silver selenide-based composite film for thermoelectric devices

Yao Lu,^a Yang Qiu,^b Kefeng Cai,^{*a} Yufei Ding,^a Mengdi Wang,^c Cong Jiang,^a
Qin Yao,^c Changjun Huang,^a Lidong Chen^{*cd} and Jiaqing He^{*b}

Correction for 'Ultrahigh power factor and flexible silver selenide-based composite film for thermoelectric devices' by Yao Lu et al., *Energy Environ. Sci.*, 2020, DOI: 10.1039/c9ee01609k.

Following publication of this article the authors noted that the thickness of the films, which affects the electrical conductivity, is affected by the synthesis temperature of the silver selenide-based nanowires. When the experiments were repeated the room temperature was around 10 °C higher than when the original experiments were performed and this may influence the quality of the synthesized nanowires and finally the properties of the films.

The experiments have been redone in order to obtain repeatable data which differs slightly to that published in the original article. The authors wish to replace Fig. 2, 4 and S4 with new figures as follows. This correction does not affect any other data or change any of the scientific conclusions in the article.

Fig. 2 should be replaced by:

Supplementary Fig. S4 should be replaced by:

Fig. S4 A typical fractured surface SEM image of the composite films.

The authors also wish to replace the data related to Fig. 2, 4 and S4 in the manuscript. For example, " $2231.5 \mu\text{W m}^{-1} \text{K}^{-2}$ " should read as " $1593.9 \mu\text{W m}^{-1} \text{K}^{-2}$ " and " $zT \sim 0.5$ at 300 K" should read as " $zT \sim 0.4$ at 300 K".

The Royal Society of Chemistry apologises for these errors and any consequent inconvenience to authors and readers.

Fig. R1 Correction on *Energy Environ. Sci.*, 2020, 13, 1240 by Cai et al..

2. In addition there are still multiple grammatical errors present despite assurances that the manuscript has been thoroughly proof-read. This is unacceptable after two rounds of revision and I cannot recommend publication for this reason.

Response: Sorry for our carelessness. In order to correct the grammatical errors and improve the English language, we used the Nature Research Editing Service, with the Editing Certificate shown below. We believe now the manuscript should be suitable for publication.

SPRINGER NATURE
Author Services Editing Certificate

This document certifies that the manuscript

Modulation of the morphotropic phase boundaries of high-performance ductile thermoelectric materials

prepared by the authors

Jiasheng Liang, Jin Liu, Pengfei Qiu, Chen Ming, Zhengyang Zhou, Zhiqiang Gao ,
Kunpeng Zhao, Lidong Chen, and Xun Shi

was edited for proper English language, grammar, punctuation, spelling, and overall style
by one or more of the highly qualified native English speaking editors at SNAS.

This certificate was issued on **November 20, 2023** and may be verified
on the SNAS website using the verification code **EA01-E059-B1D3-6F6A-9E98** .

Neither the research content nor the authors' intentions were altered in any way during the editing process. Documents receiving this certification should be English-ready for publication; however, the author has the ability to accept or reject our suggestions and changes. To verify the final SNAS edited version, please visit our verification page at secure.authorservices.springernature.com/certificate/verify.
If you have any questions or concerns about this edited document, please contact SNAS at support@as.springernature.com.

List of changes

1. The grammatical errors were corrected by using the Nature Research Editing Service.